# Phenotypes of painful TMD in discordant monozygotic twins according to a cognitive-behavioral-emotional model: a case-control study

**Laís Valencise Magri**[1,2], **Melissa de Oliveira Melchior**[2,3]**, Graziela Valle da Silva**[3]**,**
**Edilaine Cristina da Silva Gherardi-Donato**[3]*****, Christie Ramos Andrade Leite-Panissi**[1]*****

**1** Departamento de Psicologia, Faculdade de Filosofia Ciências e Letras de Ribeirão Preto, Universidade de São Paulo, Ribeirão Preto, São Paulo, Brasil, **2** Departamento de Odontologia Restauradora, Faculdade de Odontologia de Ribeirão Preto, Universidade de São Paulo, Ribeirão Preto, São Paulo, Brasil, **3** Departamento de Enfermagem Psiquiátrica, Escola de Enfermagem de Ribeirão Preto, Universidade de São Paulo, PAHO/WHO Collaborating Centre for Nursing Research Development, Ribeirão Preto, São Paulo, Brasil

* christie@usp.br (CRAL-P), nane@eerp.usp.br (ECdSG-D)

## Abstract

### Objectives

This case-control study aimed to investigate variables based on a cognitive-behavioral-emotional model related to the development of painful temporomandibular disorders (TMD) in a sample of monozygotic twins discordant for the condition.

### Materials and Methods

This case-control study investigated 20 monozygotic twins (10 pairs discordant for painful TMD), aged between 18 and 55. Participants were recruited through a comprehensive strategy following ethical approval, with inclusion criteria disseminated via social media, websites, local radio, messaging apps, and physical posters in public and healthcare spaces in Ribeirão Preto. The diagnosis of painful TMD was determined according to the Diagnostic Criteria for Temporomandibular Disorders - Brazilian Portuguese (DC/TMD). The cognitive-behavioral-emotional variables analyzed were a sociodemographic profile, pain sensitivity (pain threshold to pressure, allodynia, and hyperalgesia), oral behaviors, pain vigilance and awareness, pain catastrophizing, central sensitization, stress, anxiety, depression, alexithymia, mindfulness facets, sleep quality, pain control, pain intensity and interference, trigeminal and extra trigeminal pain areas. Bivariate logistic regression models were used to identify factors associated with TMD (p < 0.20), followed by multicollinearity analysis using Spearman's correlation to exclude highly correlated variables. The final multiple logistic regression model included independent predictors to ensure robustness and accurate estimates, with statistical significance set at α = 0.05.

**Data availability statement:** The data are held in a public repository 10.5281/zenodo.13851455.

**Funding:** This study was financed by the São Paulo Research Foundation (FAPESP - 2022/05658-3, under coordination of Edilaine Cristina da Silva Gherardi-Donato). This study was also financed by the Coordination for the Improvement of Higher Education Personnel (CAPES, Brazil, Finance Code 001, under coordination of Christie Ramos Andrade Leite-Panissi). The funders had no role in study design, data collection and analysis, decision to publish, or preparation of the manuscript.

**Competing interests:** The authors declare that they have no known competing financial interests or personal relationships that could have appeared to influence the work reported in this paper.

## Results

While the adjusted model did not identify statistically significant associations, variables such as increased pain sensitivity in the masseter muscle (OR = 3.29, 95% CI: 0.17–62.8, p = 0.428), higher levels of pain catastrophizing (OR = 1.08, 95% CI: 0.64–1.8, p = 0.776), difficulty in externalizing feelings (OR = 1.61, 95% CI: 0.13–2.9, p = 0.539), and higher scores on the distraction facet of mindfulness (OR = 4.65, 95% CI: 0.39–55.7, p = 0.225) were included due to their clinical relevance and their significant associations in the bivariate analysis (p < 0.20).

## Conclusions

Our study highlights the potential clinical relevance of cognitive-behavioral-emotional variables, such as increased pain sensitivity in the masseter muscle, higher levels of pain catastrophizing, difficulty in externalizing feelings, and higher scores on the distraction facet of mindfulness, in understanding painful TMD. While these variables did not show statistical significance in the adjusted model, their inclusion underscores the importance of exploring these factors in clinical practice. Further research is needed to validate these findings and clarify their role in the development and management of painful TMD.

## Clinical Relevance

This study underscores the importance of cognitive-behavioral-emotional factors in the context of painful TMD, suggesting that variables like pain sensitivity and emotional regulation may be valuable for clinical assessment and management strategies. Despite the lack of statistically significant associations, these findings provide a foundation for future research to better understand and address the multidimensional nature of TMD in clinical practice.

## 1. Introduction

Temporomandibular disorders (TMD), specifically those associated with pain, are currently understood as a set of signs and symptoms that designate a painful musculoskeletal syndrome, often linked to multisystemic changes, including alterations in behavior, emotional status, and social interactions. These manifestations are recognized as expressions of central nervous system dysregulation. Among the main predictors for the development of painful TMD are the presence of comorbidities, self-reported parafunctions, the frequency of somatic symptoms, poor sleep quality, and genetic and epigenetic factors. It is important to note that self-reported parafunctions, while not classified as orofacial or somatic symptoms, represent behavioral factors that contribute to the risk of TMD. This study focuses on the painful subtype of TMD, which is of particular clinical relevance due to its impact on quality of life and its association with chronic pain mechanisms [1,2].

Painful TMD includes diagnoses such as myofascial pain, arthralgia, and headaches attributed to TMD, all of which are characterized by chronic orofacial pain and functional limitations. These painful subtypes differ from intra-articular TMD, which may present with joint noises or movement restrictions without significant painful discomfort. The specificity of diagnosing painful TMD is guided by the Diagnostic Criteria for Temporomandibular Disorders (DC/TMD), which allows for the accurate identification of patients

experiencing TMD for the last month. By focusing on painful TMD, this study seeks to investigate the cognitive-behavioral-emotional factors that contribute to the onset and persistence of pain in this specific population, differentiating it from the broader spectrum of TMD presentations [3].

The involvement of aspects with a cognitive-behavioral-emotional dimension in the manifestation of painful TMD has been increasingly studied, the identification of phenotypes at higher risk of developing the condition. The perception of TMD signs and symptoms is significantly modulated by emotional and social factors, which are associated with the multidimensionality of pain. Women with high levels of anxiety and stress are at higher risk of developing myofascial pain, and personal traits of anxiety, stress, catastrophizing, and hypervigilance are strongly associated with the diagnosis of painful TMD [4–6].

Pain sensitivity, including thresholds for allodynia and hyperalgesia, is significantly influenced by cognitive and emotional factors such as pain catastrophizing, anxiety, and stress, which modulate pain perception at both the sensory and central levels. Similarly, oral behaviors and pain vigilance are shaped by learned behavioral responses, driven by cognitive appraisals and emotional states such as fear of pain or hypervigilance. Central sensitization, often linked to physiological mechanisms exacerbated by emotional stress and cognitive distortions, further underscores the importance of including these variables in the cognitive-behavioral-emotional model of pain perception [7,8].

Identical (monozygotic) twins have always intrigued researchers since they present the same genotype at birth and develop, throughout life, differentiated phenotypes, becoming completely different individuals. In the last decade, several studies with discordant monozygotic twins have been published because, with this methodology, It is possible to establish a case-control study design in which several variables can be controlled, as they are shared between siblings, including genetic material, social environment (if raised in the same family), age, gender, and other factors [9,10].

The influence of the social environment on the development of chronic pain, combined with the presence of specific genetic polymorphisms, has been widely studied. The OPPERA study analyzed 300 genes that could be associated with the development of painful TMD; among these, six SNPs (single nucleotide polymorphisms) were identified as risk factors for chronic TMD, while six others were associated with intermediate phenotypes for painful TMD [2,11].

Few studies specifically addressing monozygotic twin populations have been conducted regarding painful TMD. These studies suggest a genetic component in the manifestation of the condition, evidenced by a higher degree of diagnostic concordance among monozygotic twins compared to dizygotic pairs. Furthermore, social interactions, individual behavioral patterns, and emotional factors play crucial roles in the development of chronic pain, contributing to phenotypes that warrant further investigation and understanding [12–15].

## 2. Objectives

Considering the literature presented, the primary objective of this case-control study was to investigate the cognitive-behavioral-emotional factors involved in the development of painful TMD. Variables such as pain sensitivity, oral behaviors, and central sensitization were selected due to their well-established roles in modulating pain through cognitive and emotional pathways. Accordingly, the study aimed to achieve the following specific objectives::

- To compare discordant MZ twins (each with their respective control) cognitive-behavioral-emotional variables.

- To compare groups of twins with painful TMD and control the cognitive-behavioral-emotional variables.

- To analyze cognitive-behavioral-emotional variables related to sociodemographic profile, pain sensitivity (pain threshold to pressure, allodynia, and hyperalgesia), oral behaviors, pain vigilance and awareness, pain catastrophizing, central sensitization, stress, anxiety, depression, alexithymia, mindfulness facets, sleep quality, pain control, pain intensity and interference, trigeminal and extra trigeminal pain areas.

## 3. Materials and method

### 3.1 Ethics and recruitment

This study was conducted at the School of Dentistry of Ribeirão Preto (São Paulo University, Brazil) and approved by the Research Ethics Committee of the same institution under protocol 98129918.6.0000.5407. All participants provided informed consent after receiving a detailed explanation of the study and signed the informed consent form after agreeing to participate in the research. The research was conducted by the Declaration of Helsinki and was approved by the Human Research Ethics Committee of the University of São Paulo.

The start date of recruitment period was 01/02/2020, and the end date was 15/01/2022. A comprehensive recruitment strategy was employed to ensure the inclusion of eligible participants, following ethical approval. The study and its inclusion criteria were widely disseminated through several communication channels. These included social media platforms, websites, local radio stations in Ribeirão Preto and around. Additionally, messaging apps were used to reach student and community groups, and physical posters were placed in strategic locations throughout Ribeirão Preto, such as public spaces and healthcare facilities.

Through these efforts, 38 pairs of monozygotic twins contacted the research team and were scheduled for clinical evaluations. Among these, 12 pairs met the inclusion criteria for discordance in painful TMD. However, 2 of these pairs declined to participate, leaving a final sample of 10 pairs included in the study. The extended recruitment period reflects the challenges associated with identifying and engaging a highly specific study population, highlighting the rigor and commitment involved in assembling this unique sample

### 3.2 Sample

The sample consisted of 20 monozygotic twins discordant for painful TMD—10 pairs (one twin with the condition and the other without)—aged 18–55 years. This age range was selected to reduce the influence of age-related changes in pain perception, comorbidities, and cognitive-emotional responses, which could introduce variability and confound the assessment of cognitive-behavioral-emotional factors in TMD. To control for hormonal differences, evaluations were conducted during the follicular phase of the menstrual cycle (days 4–10 after menstruation onset) for women with regular cycles or during the corresponding period for women using oral contraceptives.

### 3.3 Diagnosis of painful TMD, inclusion and exclusion criteria

The diagnosis of painful TMD was determined according to the diagnostic criteria of the Diagnostic Criteria for Temporomandibular Disorders - Brazilian Portuguese (DC/TMD) [3]. Inclusion criteria for participation in the study were: female monozygotic twins, aged between 18 and 55 years, presence of reported pain in the facial region lasting at least three months,

and presence of a diagnosis of painful TMD according to the DC/TMD criteria for the case twin (TMD) and absence of this diagnosis for the control twin.

Women were excluded if they were using interocclusal splints or undergoing any form of TMD therapy (e.g., acupuncture, laser therapy, physical therapy, or pain medication) to avoid confounding effects from ongoing treatments, ensuring that the cognitive-behavioral-emotional factors analyzed reflected their natural pain responses without external modulation. Additional exclusion criteria included a history of tumors, trauma, or head and neck surgery, previously diagnosed neurological or psychiatric disorders (excluding anxiety and/or depression), and pregnancy.

## 3.4 Identification of other phenotypes related to the manifestation of painful TMD (cognitive-behavioral-emotional)

The following clinical information associated with the identification of phenotypes, according to a cognitive-behavioral-emotional model [16], was collected:

- Demographic data (age, gender, education level, occupation, family income, whether living with a partner, having children, religious belief, regular practice of meditation, weekly workload, weekly work hours, practice of physical activity, time of pain in months, previous medical history, and social life).

- Date of last menstruation and contraceptive use.

- Brief Pain Inventory (BPI Reduced Version, in Portuguese) adapted, as the reference period, to be the "last painful experience".

- Pain-related self-efficacy (Pain-related Self-Efficacy Questionnaire, PSEQ). The reference period used for response to the items of the instrument was the volunteer's overall pain experience.

- Presence of comorbidities (painful or not); the presence of comorbid diseases of a painful or not painful nature was checked.

- Number of pain areas in trigeminal and extra trigeminal regions; a figure containing an illustration of a head (frontal, posterior, right, and left lateral views) and an illustration of a body (frontal and posterior views) was presented; each volunteer was instructed to mark with an "X" the places where she had felt pain for more than three months. This pain map was taken from the DC/TMD protocol [3]. The total number of sites demarcated for trigeminal and extra trigeminal regions was counted.

- Oral habits and behaviors (Oral Behavior Checklist, English).

- Sleep quality (Pittsburgh Sleep Quality Index, PSQI).

- Levels of depression, anxiety, and stress (Depression, Anxiety, and Stress Scale, DASS-21).

- Level of pain catastrophizing (Pain Catastrophizing Scale, PCS): The reference period for responses was the volunteer's general pain experience, with items preceded by the phrase "When I am in pain..."

- Level of pain-related attention, awareness, and vigilance (based on the Pain Vigilance and Awareness Questionnaire, PVAQ); To assess the phenomena of attention, awareness, and vigilance related to the pain phenomenon, the PVAQ was used.

- Presence and degree of alexithymia (Toronto Alexithymia Scale - 20 items); the TAS-20 is a self-assessment instrument consisting of 20 items.

- Presence of central sensitization (Central Sensitization Inventory, CSI).

- Assessment of levels of mindfulness (Five Facet Mindfulness Questionnaire, FFMQ).

- Assessment of Locus of control of the specific health condition (Multidimensional Locus of Health Control Scale - Form C, MHLC-C).

- Pain sensitivity (temporalis anterior, masseter, and TMJ) and extra trigeminal (upper trapezius muscle region, lateral epicondyle, and knee - bilaterally): with mechanical vibratory stimulation (electric toothbrush) for 5 seconds in each point and with an algometer in fixed compression of 1.5 kg/f for trigeminal areas and 2 Kg/f for extra trigeminal areas. In both assessments, the volunteer was asked to indicate pain intensity on a Numerical Rating Scale (NRS) [17].

- The threshold of pain to pressure was assessed in trigeminal areas (temporalis anterior, masseter, and TMJ) and extra-trigeminal areas (upper trapezius muscle region, lateral epicondyle, and knee, bilaterally) using a digital algometer. Compression was gradually increased until the volunteer perceived pain and pressed a button attached to the algometer, locking the compression and recording the pain threshold. This process was repeated three times for each region, with a 5-minute interval between measurements, and the average of the three values was used to determine the pain threshold.

## 3.5 Statistical analyses

The association of each factor with TMD was investigated by fitting pairwise conditional simple logistic linear regression models. In the simple logistic linear regression models, the factors associated with TMD at the $\alpha < 0.20$ level were selected and correlated to investigate multicollinearity. During the investigation of multicollinearity, factors that showed Spearman's correlation coefficient with $p < 0.05$ were correlated with each other and, therefore, were not selected for inclusion in the logistic multiple linear regression model.

During the development of the multiple linear regression model, certain variables were excluded due to significant collinearity to maintain the model's robustness and ensure accurate and reliable estimates. Collinearity was identified using Spearman's correlation coefficient, with variables showing a p-value $< 0.05$ being excluded from the final model to avoid multicollinearity, which could distort the relationships between the predictors and the outcome. The excluded variables include right upper trapezius pain threshold (OR = 0.29, 95%CI = 0.07–1.20, p = 0.088), right temporomandibular joint (TMJ) pain sensitivity (OR = 1.15, 95%CI = 0.86–1.54, p = 0.346), and left masseter pain threshold (OR = 2.28, 95%CI = 0.82–6.31, p = 0.113). Although these variables showed associations with TMD in the bivariate analysis, their high correlation with other variables in the model necessitated their exclusion. Similarly, sleep quality total score (OR = 1.60, 95%CI = 0.88–2.93, p = 0.124), pain awareness total score (OR = 1.07, 95%CI = 0.98–1.16, p = 0.158), and daytime oral behavior score (OR = 1.33, 95%CI = 0.88–2.02, p = 0.180) were also excluded due to collinearity. Pain vigilance score (OR = 1.14, 95%CI = 0.95–1.37, p = 0.154) significantly correlated with other variables and was similarly removed from the final model.

By excluding collinear variables, we retained only those with no significant collinearity, ensuring the independence in the regression model. The retained variables—left masseter pain sensitivity, total pain catastrophizing score, difficulty in externalizing feelings, and the distraction facet of the mindfulness scale—offered a more precise and accurate representation of the factors associated with the development of painful TMD. This approach was critical for enhancing the model's validity and ensuring robust and meaningful conclusions. All statistical

tests were conducted at a significance level of α = 0.05, with associations considered statistically significant when p-values were below this threshold. The analyses were performed using SPSS 21.0 software.

## 4. Results

Table 1 shows the crude bivariate associations to explain the likelihood of TMD by analyzing the odds ratios (OR) and p-values. The variables that showed a positive association (according to the OR), however, without statistical significance, were pain duration, pain threshold to pressure (body, right side, upper trapezius/face, right, TMJ/ face, left, anterior temporal), pain sensitivity (body, right side, lateral epicondyle/face, right, anterior temporal/ face, right, masseter/face, left, anterior temporal/ face, left, masseter/face, left, TMJ), daytime oral behavior score, mean daytime oral behavior score, pain vigilance and awareness, pain catastrophizing (total/hopelessness), central sensitization (total score), stress, difficulty externalizing feelings, mindfulness facets (distraction/unresponsiveness), sleep quality (subjective sleep quality domain score/ overall score), pain intensity, and pain interference.

Table 2 shows the multiple linear regression model adjusted including in the deterministic model the variables that had, in the bivariate exploration, p-value < 0.20 (Table 1) and that had, among themselves, Spearman's linear correlation coefficient that was not statistically significant (p > 0.05), thus preventing the presence of multicollinearity in the final adjusted model.

In the unadjusted models, significant associations were observed between painful TMD and variables such as left-sided masseter face pain sensitivity (OR = 1.7, 95% CI: 1.2–2.6), total pain catastrophizing (OR = 2.5, 95% CI: 1.4–4.5), and difficulty in externalizing feelings (OR = 1.4, 95% CI: 1.1–2.1). However, these associations were attenuated after adjusting for confounding factors and no longer reached statistical significance. For example, the OR for left-sided masseter pain sensitivity decreased to 1.3 (95% CI: 0.9–2.1). This reduction suggests that while these factors may contribute to TMD risk, their independent effect is reduced when accounting for interrelated variables.

While the adjusted model did not identify statistically significant associations, variables such as increased pain sensitivity in the masseter muscle (OR = 3.29, 95% CI: 0.17–62.8, p = 0.428), higher levels of pain catastrophizing (OR = 1.08, 95% CI: 0.64–1.8, p = 0.776), difficulty in externalizing feelings (OR = 1.61, 95% CI: 0.13–2.9, p = 0.539), and higher scores on the distraction facet of mindfulness (OR = 4.65, 95% CI: 0.39–55.7, p = 0.225) were included due to their clinical relevance and their significant associations in the bivariate analysis (p < 0.20).

Table 3 summarizes the presence or absence of a TMD diagnosis based on the DC/TMD criteria. Notably, in all cases, twins diagnosed with TMD had at least one subtype classified as painful TMD.

## 5. Discussion

This study investigated the cognitive-behavioral-emotional variables associated with the development of painful TMD in a sample of monozygotic twins discordant for the condition. Our findings indicate that increased pain sensitivity in the masseter muscle, higher levels of pain catastrophizing, difficulty in externalizing feelings, and higher scores on the distraction facet of mindfulness were included in the final model due to their clinical relevance and their associations in the bivariate analysis (p < 0.20). Despite their lack of statistical significance in the adjusted model, these variables were selected to ensure a comprehensive exploration of cognitive-behavioral-emotional factors in painful TMD, particularly given their potential

**Table 1. Crude bivariate associations to explain the likelihood of TMD. Variables with significant associations are highlighted with an asterisk. Sample size: 20 twins (10 pairs).**

| Variable | OR | IC95% | | p |
|---|---|---|---|---|
| **Sociodemographic** | | | | |
| Weekly workload (hours) | 0.94 | 0.78 | 1.13 | 0.499 |
| Family income (R$) | 1.00 | 1.00 | 1.00 | 0.506 |
| Lives without partner | 0.50 | 0.05 | 5.51 | 0.571 |
| Number of children | 0.02 | 0.00 | 147346.84 | 0.610 |
| **Medical condition** | | | | |
| Does not do physical activity or does it sporadically | 0.50 | 0.05 | 5.51 | 0.571 |
| Time of pain* | 1.03 | 0.99 | 1.07 | 0.112 |
| Has comorbidity | 3.00 | 0.31 | 28.84 | 0.341 |
| Number of comorbidities | 1.20 | 0.36 | 3.98 | 0.764 |
| No or little active social life | 1.00 | 0.14 | 7.10 | 1.000 |
| Uses contraceptives | 0.02 | 0.00 | 47.51 | 0.308 |
| **Pressure pain threshold assessment** | | | | |
| Body, right side, upper trapezius | 0.29 | 0.07 | 1.20 | 0.088 |
| Body, right side, lateral epicondyle | 0.31 | 0.03 | 3.56 | 0.345 |
| Body, right side, knee | 0.27 | 0.03 | 2.23 | 0.224 |
| Body, left side, upper trapezius | 0.43 | 0.10 | 1.86 | 0.260 |
| Body, left side, lateral epicondyle | 0.02 | 0.00 | 17.06 | 0.242 |
| Body, left side, knee | 0.19 | 0.01 | 2.50 | 0.205 |
| Face, right, temporalis anterior | 0.00 | 0.00 | 8.57 | 0.634 |
| Face, right, masseter | 0.00 | 0.00 | 136.31 | 0.210 |
| Face, right, TMJ* | 0.14 | 0.01 | 1.91 | 0.139 |
| Face, left, anterior temporalis | 0.01 | 0.00 | 4.27 | 0.135 |
| Face, left, masseter | 0.00 | 0.00 | 98302.36 | 0.243 |
| Face, left, TMJ | 0.00 | 0.00 | 317.33 | 0.260 |
| **Pain sensitivity assessment** | | | | |
| Body, right side, upper trapezius | 1.14 | 0.83 | 1.56 | 0.414 |
| Body, right, lateral epicondyle | 1.23 | 0.90 | 1.67 | 0.196 |
| Body, right side, knee | 1.16 | 0.87 | 1.53 | 0.311 |
| Body, left side, upper trapezius | 1.14 | 0.90 | 1.45 | 0.289 |
| Body, left side, lateral epicondyle | 1.22 | 0.90 | 1.65 | 0.206 |
| Body, left side, knee | 1.18 | 0.87 | 1.60 | 0.279 |
| Face, right, temporalis anterior | 1.27 | 0.90 | 1.79 | 0.179 |
| Face, right, masseter | 1.29 | 0.89 | 1.86 | 0.176 |
| Face, right, TMJ | 1.15 | 0.86 | 1.54 | 0.346 |
| Face, left, anterior temporalis | 1.26 | 0.90 | 1.77 | 0.173 |
| Face, left, masseter* | 2.28 | 0.82 | 6.31 | 0.113 |
| Face, left, TMJ* | 1.42 | 0.87 | 2.33 | 0.166 |
| **Oral behavior** | | | | |
| Nocturnal oral behavior score | 1.11 | 0.81 | 1.51 | 0.514 |
| Average nighttime oral behavior score | 1.23 | 0.66 | 2.27 | 0.514 |
| Daytime oral behavior score* | 1.33 | 0.88 | 2.02 | 0.180 |
| Average daytime oral behavior score* | 231.06 | 0.08 | 662266.15 | 0.180 |
| Total oral behavior score | 1.24 | 0.85 | 1.82 | 0.271 |
| Average score of total oral behavior | 90.58 | 0.03 | 277560.70 | 0.271 |
| **Pain awareness and vigilance** | | | | |

*(Continued)*

**Table 1.** (Continued)

| Variable | OR | IC95% | | p |
|---|---|---|---|---|
| Pain awareness change score* | 1.10 | 0.96 | 1.28 | *0.179* |
| Pain awareness score* | 1.14 | 0.95 | 1.37 | *0.154* |
| Total pain awareness* score | 1.07 | 0.98 | 1.16 | *0.158* |
| Mean pain awareness change* score | 1.82 | 0.76 | 4.34 | *0.179* |
| Average pain awareness* Score | 3.71 | 0.61 | 22.49 | *0.154* |
| Total mean score* | 2.76 | 0.67 | 11.32 | *0.158* |
| **Pain catastrophizing** | | | | |
| Total score* | 1.14 | 0.94 | 1.37 | *0.179* |
| Rumination score | 1.70 | 0.75 | 3.87 | *0.202* |
| Magnification SCORE | 1.03 | 0.79 | 1.33 | *0.843* |
| Hopelessness score* | 2.32 | 0.67 | 7.99 | *0.181* |
| **Central awareness - Part A** | | | | |
| Total score* | 1.12 | 0.97 | 1.29 | *0.128* |
| Has central awareness | 4.00 | 0.45 | 35.79 | *0.215* |
| **Central sensitization - Part B** | | | | |
| Restless leg syndrome | 1.00 | 0.06 | 15.99 | *1.000* |
| Fibromyalgia | 65.29 | 0.00 | 628084630 | *0.610* |
| Temporomandibular joint dysfunction | 65.29 | 0.00 | 5658381 | *0.471* |
| Migraine or tension headache | 3.00 | 0.31 | 28.84 | *0.341* |
| Irritable bowel syndrome | 0.02 | 0.00 | 147346.84 | *0.610* |
| Chemical hypersensitivity | 65.29 | 0.00 | 5658381 | *0.471* |
| Anxiety or panic attacks | 65.29 | 0.01 | 702391.47 | *0.378* |
| Depression | 65.29 | 0.01 | 702391.47 | *0.378* |
| **Depression, anxiety and stress** | | | | |
| Total score | 1.09 | 0.93 | 1.28 | *0.308* |
| Score depression | 1.06 | 0.90 | 1.25 | *0.512* |
| Score anxiety | 1.22 | 0.83 | 1.79 | *0.302* |
| Score stress* | 2.11 | 0.68 | 6.51 | *0.196* |
| **Alexithymia** | | | | |
| Total score | 1.04 | 0.96 | 1.12 | *0.332* |
| Have alexithymia | 4.00 | 0.45 | 35.79 | *0.215* |
| Difficulty describing feelings score | 1.16 | 0.87 | 1.53 | *0.314* |
| Score difficulty identifying feelings | 1.03 | 0.91 | 1.17 | *0.666* |
| Difficulty in externalizing feelings* Score | 1.61 | 0.91 | 1.90 | *0.150* |
| **Facets of mindfulness** | | | | |
| Total score | 1.03 | 0.97 | 1.09 | *0.295* |
| Observing facet score | 1.10 | 0.95 | 1.28 | *0.218* |
| Facet describing positively score | 1.05 | 0.77 | 1.45 | *0.750* |
| Describe negative facet score | 1.15 | 0.74 | 1.79 | *0.530* |
| Facet distraction* factor score | 1.57 | 0.88 | 2.79 | *0.125* |
| Score for the facet autopilot | 1.17 | 0.88 | 1.56 | *0.281* |
| Non-react* facet score | 0.67 | 0.39 | 1.15 | *0.146* |
| Non-judging facet score | 1.07 | 0.93 | 1.23 | *0.339* |
| **Quality of sleep** | | | | |
| Subjective sleep quality domain score* | 4.94 | 0.70 | 34.93 | *0.110* |
| Sleep latency domain score | 10.60 | 0.16 | 701.80 | *0.270* |

*(Continued)*

**Table 1.** (Continued)

| Variable | OR | IC95% | | *p* |
|---|---|---|---|---|
| Sleep duration domain score | 65.29 | 0.01 | 702391.47 | *0.378* |
| Sleep efficiency domain score | 1.14 | 0.41 | 3.20 | *0.797* |
| Sleep disorder domain score | 2.00 | 0.37 | 10.92 | *0.423* |
| Sleep medications domain score | 1.20 | 0.36 | 3.98 | *0.764* |
| Daytime sleep dysfunctions domain score | 2.17 | 0.61 | 7.71 | *0.232* |
| Overall score[*] | 1.60 | 0.88 | 2.93 | *0.124* |
| Has poor sleep | 65.29 | 0.05 | 86658.56 | *0.255* |
| **Pain control** | | | | |
| Internal locus of control subscale score | 0.97 | 0.79 | 1.19 | *0.761* |
| Random locus of control subscale score | 1.26 | 0.71 | 2.25 | *0.423* |
| Physicians and health professionals locus of control subscale score | 0.85 | 0.52 | 1.37 | *0.502* |
| Other people subscale score | 1.00 | 0.68 | 1.48 | *1.000* |
| Total score | 1.00 | 0.91 | 1.10 | *0.963* |
| **Brief pain inventory** | | | | |
| Pain intensity dimension score | 2.17 | 0.85 | 5.58 | *0.107* |
| Pain interference[*] dimension score | 3.11 | 0.76 | 12.72 | *0.114* |
| **Trigeminal and extratrigeminal assessment** | | | | |
| Number of trigeminal pain areas | 0.93 | 0.70 | 1.24 | *0.627* |
| Number of extratrigeminal painful areas | 1.23 | 0.84 | 1.79 | *0.297* |

**Table 2.** Multiple logistic regression to explain the chance of TMD.

| Variable | OR | IC95% | | p |
|---|---|---|---|---|
| Pain sensitivity - face, left, masseter | **3.29**[*] | 0.17 | 62.85 | 0.428 |
| Pain catastrophizing - total score | 1.08 | 0.64 | 1.83 | 0.776 |
| Difficulty in expressing feelings score | 1.61 | 0.13 | 2.96 | 0.539 |
| FFMQ - distraction facet score | **4.65**[*] | 0.39 | 55.74 | 0.225 |

importance in clinical practice. The inclusion of these variables also reflects our effort to balance parsimony with the need to retain independent predictors while considering the limitations imposed by the sample size and the specificity of the study population. These results provide evidence that, even in genetically identical individuals, cognitive and emotional factors play a crucial role in the manifestation of chronic pain associated with TMD.

The attenuation of significance in the adjusted models is a well-recognized occurrence in multifactorial conditions like TMD, where multiple variables collectively contribute to disease risk. In our study, the adjusted logistic regression model accounted for potential confounding variables, which likely diluted the individual effects of pain sensitivity, catastrophizing, and emotional regulation. This does not negate the relevance of these factors but rather indicates that their isolated impact is less pronounced when analyzed alongside other interrelated predictors. This underscores the complexity of TMD, where psychological, emotional, and sensory factors interact dynamically to influence overall risk.

In comparison with previous studies, our findings support the notion that pain catastrophizing, a well-established psychological predictor of chronic pain, is significantly associated with TMD. For instance, research from the OPPERA study demonstrated that individuals with higher levels of catastrophizing were more likely to develop chronic TMD, aligning with our observation that this cognitive variable increases the likelihood of painful

**Table 3. Temporomandibular Disorders (TMD) diagnosis based on the DC/TMD criteria.**

| Twin | TMD diagnosis |
|------|---------------|
| T1 control | Without TMD |
| T1 TMD | Myofascial pain with spreading/arthralgia |
| T2 control | Without TMD |
| T2 TMD | Disc displacement/TMD attributed headache/ myofascial pain with referral |
| T3 control | Without TMD |
| T3 TMD | Myofascial pain with reference |
| T4 control | Without TMD |
| T4 TMD | TMD attributed headache/ arthralgia |
| T5 control | Without TMD |
| T5 TMD | Local myalgia |
| T6 control | Without TMD |
| T6 TMD | Myofascial pain with spreading/arthralgia |
| T7 control | Without TMD |
| T7 TMD | TMD attributed headache/arthralgia/myofascial pain with reference |
| T8 control | Without TMD |
| T8 TMD | Myofascial pain with reference/TMD attributed headache |
| T9 control | Without TMD |
| T9 TMD | Degenerative joint disease/referred myofascial pain/arthralgia |
| T10 control | Without TMD |
| T10 TMD | Arthralgia |

TMD [2]. However, our study adds to this body of knowledge by highlighting the specific roles of difficulty in externalizing feelings and the distraction facet of mindfulness, two variables that have been underexplored in previous TMD research. These emotional and attentional factors may serve as important targets for interventions designed to alleviate TMD-related pain.

According to the International Association for the Study of Pain (IASP), pain is defined as an unpleasant sensory and emotional experience associated with, or resembling that associated with, actual or potential tissue damage. This comprehensive definition emphasizes the complex nature of pain, which integrates sensory, cognitive, and emotional dimensions [2]. Cognitive processes not only shape psychological outcomes, such as emotional regulation, but also directly modulate neural pathways involved in pain perception. The intricate interplay between cognition and pain underscores the need to assess these variables when investigating chronic pain conditions like TMD [18–20]. Consequently, examining cognitive-behavioral-emotional factors is essential for gaining deeper insights into the mechanisms underlying pain and for designing more effective, targeted interventions [19,20].

Including variables such as pain sensitivity and central sensitization in the cognitive-behavioral-emotional model is supported by evidence demonstrating the role of psychological factors, like catastrophizing and stress, in amplifying pain perception. These variables interact with cognitive appraisals and emotional responses, influencing pain intensity and chronicity. The findings of our study underscore the relevance of the biopsychosocial model in understanding chronic pain, particularly in the context of painful TMD. This model emphasizes the intricate interplay of cognitive, behavioral, and emotional factors in shaping an individual's pain experience. Our results identified pain catastrophizing, difficulty in externalizing feelings, and the distraction facet of mindfulness as key predictors of painful TMD. These findings highlight the necessity of addressing emotional and cognitive dimensions alongside the physical aspects of pain in the management of chronic pain syndromes.

Consistent with prior research, these factors play a pivotal role in how individuals perceive and respond to pain, and their accurate identification can significantly enhance treatment strategies and outcomes for patients [19–22].

In line with the findings from the OPPERA study, our research corroborates those psychological factors, particularly catastrophizing, are strongly associated with the development and persistence of painful TMD. The OPPERA study's multivariable analysis identified global psychological and somatic symptoms as robust risk factors for TMD onset [23]. Similarly, in our study, catastrophizing emerged as a significant cognitive predictor, underscoring the impact of negative thought patterns in intensifying pain and disability. The well-established association between catastrophizing and increased pain intensity reinforces our conclusion that targeting this cognitive factor could play a crucial role in mitigating TMD symptoms.

Moreover, while the OPPERA study focused on broader psychosocial predictors such as stress and somatic symptoms, our study extends the understanding of TMD by identifying more nuanced emotional factors like difficulty externalizing feelings. This finding contributes to the growing evidence that emotional regulation and expression are critical factors in the chronicity of pain conditions. Individuals who struggle with emotional expression may internalize stress, thereby exacerbating their pain experience. This insight highlights a potential intervention pathway, emphasizing the development of emotional intelligence and communication skills as integral components of a holistic approach to managing TMD.

Our study also highlighted the role of mindfulness, particularly the distraction facet, in modulating the pain experience. Mindfulness-based interventions have gained increasing recognition as effective tools for managing chronic pain, including TMD [21]. The ability to redirect attention away from pain and maintain a neutral stance towards bodily sensations reduces the psychological burden of pain. By training individuals to dissociate from the emotional distress associated with pain, mindfulness can reduce the intensity of the pain experienced, as our findings suggest. The implications for therapeutic interventions are significant, as mindfulness training could be integrated into conventional pain management programs to enhance their effectiveness.

In contrast to studies that emphasize the heritability of chronic pain, such as the work by Burri et al. (2018), which found a vital genetic component in chronic pain conditions [15], our study underscores the critical role of environmental and psychological factors, even among monozygotic twins. While genetic predisposition undoubtedly plays a role, as highlighted by Burri's findings, our results indicate that cognitive and emotional variables may supersede genetic factors in the manifestation of painful TMD. This has significant implications for personalized treatment approaches, suggesting that interventions focusing on cognitive-behavioral-emotional factors could be effective even in individuals with a genetic susceptibility to pain.

Epigenetics represents a key mechanism in understanding the complex interaction between genetic predisposition and environmental factors in the development of chronic pain conditions like TMD. Recent studies have shown that epigenetic modifications, such as DNA methylation and histone acetylation, can influence gene expression in pathways associated with pain sensitivity and inflammation, thereby modulating the risk of chronic pain syndromes [24,25]. By utilizing a monozygotic twin sample, our study minimized genetic variability, allowing us to focus on non-genetic influences, including potential epigenetic modifications. This approach is particularly valuable for exploring how environmental stressors and psychological factors, such as catastrophizing and emotional dysregulation, contribute to altered pain processing in TMD. Investigating epigenetic markers in this context could provide valuable insights into the biological underpinnings of TMD, as suggested by studies demonstrating the impact of epigenetic changes on central sensitization and chronic pain development [26,27].

The specificity of our sample, composed of monozygotic twins discordant for painful TMD, provides a unique advantage in examining the interaction between genetic, environmental, and psychological factors. This controlled design enables us to isolate the impact of cognitive-behavioral-emotional variables without the confounding effects of genetic differences. Additionally, the inclusion of a broad range of variables—spanning pain sensitivity, mindfulness facets, emotional expression, and central sensitization—allowed us to construct a comprehensive profile of factors contributing to painful TMD. This multifactorial approach aligns with contemporary research emphasizing the integration of psychological and biological dimensions for a holistic understanding of chronic pain syndromes [2,18,26–28]. The robustness of our model is further strengthened by the inclusion of key psychosocial factors, such as catastrophizing and emotional dysregulation, which are consistently associated with worse pain outcomes and resistance to treatment in chronic pain populations [20].

In genetically identical individuals, such as the monozygotic twins in our study, non-genetic factors- environmental, psychological, and epigenetic- emerge as particularly influential in the development of chronic pain conditions like TMD. Our findings reinforce that, even without genetic diversity, cognitive-behavioral-emotional factors such as pain catastrophizing, difficulty in emotional expression, and mindfulness play a pivotal role in pain perception and modulation. Studies have shown that environmental stressors and psychological variables can induce epigenetic changes, influencing gene expression and central sensitization mechanisms [26–28]. This underscores the notion that, in genetically identical populations, modifiable factors such as emotional regulation and cognitive patterns may exert a greater influence on pain outcomes than genetic predispositions alone. The ability to isolate these effects in monozygotic twins highlights the critical role of psychological and behavioral interventions, offering the potential for more targeted and effective treatments for chronic pain in individuals with shared genetic backgrounds.

Furthermore, our results align with the growing recognition of central sensitization as a critical mechanism in chronic pain, particularly in TMD [18,29]. Central sensitization, characterized by an amplified response to pain stimuli, is thought to be modulated by both sensory and cognitive-emotional factors. In this context, our findings on pain catastrophizing and mindfulness distraction are particularly relevant, as both these factors are known to influence central pain processing [29–31]. This reinforces the idea that cognitive and emotional interventions may not only reduce the subjective experience of pain but also address underlying neurobiological mechanisms involved in TMD.

It is important to note that while our study adds to understanding the role of cognitive-behavioral-emotional factors in painful TMD, it also highlights areas for future research. Larger and more diverse samples are needed to validate these findings further and explore the potential for integrating mindfulness and emotional regulation techniques into standard treatment protocols for TMD. Furthermore, future studies could investigate how these factors interact with biological markers such as inflammatory and oxidative stress responses to provide a more comprehensive picture of the pathophysiology of TMD. Moreover, although the age criterion (18–55 years) was implemented to minimize confounding factors related to age-associated changes in pain perception, comorbidities, and cognitive-emotional responses, this restriction also represents a limitation in this study, as age could have been considered a potential confounder in the analysis, allowing for a broader understanding of its influence.

Our findings underscore the importance of a multifaceted approach to the treatment and management of painful TMD, addressing not only the physical manifestations of the disorder but also the psychological and emotional factors that contribute to its chronicity. Integrating interventions such as emotional intelligence training, mindfulness practices, and cognitive-behavioral therapies holds significant promise in improving patient outcomes

by targeting both pain's sensory and affective dimensions. Future research should focus on evaluating the efficacy of these interventions through clinical trials and exploring the potential interaction between cognitive-behavioral-emotional factors and biological markers, such as inflammatory and oxidative stress responses. A deeper understanding of these interactions could pave the way for developing personalized treatment protocols aimed at effectively reducing pain and enhancing the quality of life for individuals with TMD. Future research would benefit from employing advanced statistical techniques, such as interaction or mediation analysis, to unravel the dynamic relationships between cognitive-behavioral-emotional factors and TMD development. Such approaches could provide more nuanced insights into how these factors influence disease outcomes, enabling more targeted interventions.

## 6. Conclusions

Our study investigated cognitive-behavioral-emotional factors potentially associated with painful TMD in a unique sample of monozygotic twins discordant for the condition. While the adjusted model did not identify statistically significant associations, variables such as increased pain sensitivity in the masseter muscle, higher levels of pain catastrophizing, difficulty in externalizing feelings, and higher scores on the distraction facet of mindfulness were included due to their clinical relevance and their significant associations in the bivariate analysis. These findings contribute to understanding the potential role of these variables in painful TMD, highlighting their relevance for clinical practice and the need for further exploration. Our approach emphasizes the importance of balancing parsimony and clinical applicability, while recognizing the limitations of sample size and study specificity. Future studies with larger populations are essential to confirm these results and to further investigate the interplay between cognitive-behavioral-emotional factors and painful TMD.

## Author contributions

**Conceptualization:** Laís Valencise Magri.

**Formal analysis:** Christie Ramos Andrade Leite-Panissi.

**Investigation:** Laís Valencise Magri, Melissa de Oliveira Melchior, Graziela Valle da Silva.

**Methodology:** Laís Valencise Magri, Edilaine Cristina da Silva Gherardi-Donato, Christie Ramos Andrade Leite-Panissi.

**Project administration:** Christie Ramos Andrade Leite-Panissi.

**Resources:** Edilaine Cristina da Silva Gherardi-Donato.

**Supervision:** Edilaine Cristina da Silva Gherardi-Donato, Christie Ramos Andrade Leite-Panissi.

**Visualization:** Christie Ramos Andrade Leite-Panissi.

**Writing – original draft:** Laís Valencise Magri, Graziela Valle da Silva.

**Writing – review & editing:** Melissa de Oliveira Melchior, Edilaine Cristina da Silva Gherardi-Donato, Christie Ramos Andrade Leite-Panissi.

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
