## [Decision Letter · Decision Letter 0]

30 Aug 2024

PONE-D-24-27113Phenotypes of painful TMD in discordant monozygotic twins according to a cognitive-behavioral-emotional model: a case-control studyPLOS ONE

Dear Dr. Leite-Panissi,

Thank you for submitting your manuscript to PLOS ONE. After careful consideration, we feel that it has merit but does not fully meet PLOS ONE’s publication criteria as it currently stands. Therefore, we invite you to submit a revised version of the manuscript that addresses the points raised during the review process.

**ACADEMIC EDITOR: Please insert comments here and delete this placeholder text when finished.**

Dear Authors, 

Kindly review the manuscript ad make necessary changes as suggested by the reviewers. 

Kindly submit the revised version. 

Best Wishes

We look forward to receiving your revised manuscript.

Kind regards,

Kumar Chandan Srivastava, BDS, MDS, PhD, MFD RSCI, MFDS RCPS, MFDS RCSEd MDT

Academic Editor

PLOS ONE

Journal Requirements:

3. We note that your Data Availability Statement is currently as follows: "We need the journal’s help to make your data available."

Please confirm at this time whether or not your submission contains all raw data required to replicate the results of your study. Authors must share the “minimal data set” for their submission. PLOS defines the minimal data set to consist of the data required to replicate all study findings reported in the article, as well as related metadata and methods (https://journals.plos.org/plosone/s/data-availability#loc-minimal-data-set-definition ).

If your submission does not contain these data, please either upload them as Supporting Information files or deposit them to a stable, public repository and provide us with the relevant URLs, DOIs, or accession numbers. For a list of recommended repositories, please see https://journals.plos.org/plosone/s/recommended-repositories .

Additional Editor Comments :

Dear Authors,

Kindly review the manuscript ad make necessary changes as suggested by the reviewers.

Kindly submit the revised version.

Best Wishes

Reviewers' comments:

Reviewer's Responses to Questions

**Comments to the Author**

1. Is the manuscript technically sound, and do the data support the conclusions?

Reviewer #1: Yes

Reviewer #2: No

2. Has the statistical analysis been performed appropriately and rigorously? 

Reviewer #1: Yes

Reviewer #2: No

3. Have the authors made all data underlying the findings in their manuscript fully available?

Reviewer #1: Yes

Reviewer #2: Yes

4. Is the manuscript presented in an intelligible fashion and written in standard English?

Reviewer #1: Yes

Reviewer #2: No

5. Review Comments to the Author

Reviewer #1: The topic of the study is both interesting and well-defined. However, the discussion section could benefit from further refinement.

In the first paragraph of the introduction, it would be beneficial to better describe the specificity of the TMD diagnoses. For instance, the study focuses specifically on painful TMD, a point that could be highlighted earlier in the introduction.

The methodology is robust and well-detailed. The statistical analyses and results are presented clearly and concisely. It would be helpful to explicitly mention which variables were excluded due to collinearity during the regression model formation. While this is somewhat implied, providing this information would aid in the reader’s interpretation.

The discussion section is somewhat lengthy and does not thoroughly explore the study’s findings. Previous studies could be summarized more concisely and directly linked to your results.

I suggest placing greater emphasis on the aspects related to epigenetics, the specificity of your sample, and the extensive number of variables investigated to construct the model—these are significant strengths of your study.

Another aspect that could be more thoroughly discussed is the idea that in genetically identical individuals, other factors may play a more significant role than those typically identified in genetically diverse populations.

The conclusion could be more concise and clear.

*Note:* On page 12, in Table 3, the label for G7 is missing a "T."

Reviewer #2: Phenotypes of painful TMD in discordant monozygotic twins according to a cognitivebehavioral-emotional model: a case-control study

It was described that the objective of this study “was to investigate variables based on a cognitive-behavioral-emotional model related to the development of painful temporomandibular disorders (TMD) in a sample of monozygotic twins discordant for the condition.”

Please see my comments below:

1. Objective. It is important a clarification why sociodemographic profile, pain sensitivity (pain threshold to pressure, allodynia, and hyperalgesia), oral behaviors, pain vigilance, central sensitization, pain intensity and interference, trigeminal and extratrigeminal pain areas were considered cognitive-behavioral-emotional variables.

2. Method. It is important for the authors to describe the cohort from which the painful TMD patients were recruited, including how many declined participation and how many were excluded. Additionally, the rationale for including only patients aged 18-55 years needs clarification, as well as the reason for excluding older individuals. It is also unclear why women using interocclusal plates or undergoing other TMD therapies (such as acupuncture, laser therapy, physical therapy, or pain medication) were excluded. This suggests that the study may have selectively included patients with only mild pain.

3. Statistical analysis. The authors could use VIF to assess multicollinearity. Additionally, it is important to note that logistic regression is not a linear regression model. If odds ratios were estimated, then logistic regression was conducted, not linear analysis. I recommend that the authors specify the alpha level in the methods section and report p-values in the results section. It may also be beneficial for the authors to consult a statistician to review the analysis.

4. Results. The authors described an increased likelihood of TMD with rising scores in left-sided masseter face pain sensitivity, total pain catastrophizing, difficulty in externalizing feelings, and the distraction facet of the FFMQ scale. However, the adjusted model showed that none of these variables were significantly associated with painful TMD.

5. Conclusion. Therefore, I disagree with the authors' conclusion that the cognitive-behavioral-emotional model studied in this case-control study of monozygotic twins discordant for painful TMD suggests that facial pain sensitivity, pain catastrophizing, difficulty in externalizing feelings, and the distraction facet of mindfulness increase the likelihood of developing painful TMD. My conclusion is the opposite, as none of these factors were significantly associated with painful TMD.

6. Tables

Table 1 should specify the sample size included and indicate whether the analysis is crude or multivariable. It is also important to note that none of the variables were found to be associated with painful TMD.

Table 2 presents result from a multiple logistic regression analysis, not a linear regression, as indicated by the estimated odds ratios. Again, no associations were found with painful TMD.

Table 3 should include the number of participants in each group.

7. English revision is needed.

6. PLOS authors have the option to publish the peer review history of their article (what does this mean? ). If published, this will include your full peer review and any attached files.

**Do you want your identity to be public for this peer review?** For information about this choice, including consent withdrawal, please see our Privacy Policy .

Reviewer #1: No

Reviewer #2: **Yes: ** Ana Velly

---

## [Author Response · Author response to Decision Letter 1]

27 Sep 2024

Reviewer #1 Manuscript PONE-D-24-27113

General comments

The topic of the study is both interesting and well-defined. However, the discussion section could benefit from further refinement.

Response: We would like to thank you for your detailed analysis of our manuscript. Below, we present clarifications of the points raised, and the manuscript has been reformulated based on the suggestions made. We believe the revised version has been improved, and we would like a new manuscript analysis.

Specific comments

1. In the first paragraph of the introduction, it would be beneficial to better describe the specificity of the TMD diagnoses. For instance, the study focuses specifically on painful TMD, a point that could be highlighted earlier in the introduction.

Response: We appreciate the reviewer’s insightful suggestion. To address this, we have revised the first paragraph of the introduction to emphasize the specific focus on painful temporomandibular disorders (TMD) earlier in the text and added a second sentence about TMD diagnosis. We have clarified that the study specifically targets painful TMD, which allows for a more precise understanding of the population under investigation. This revision clarifies the study’s objectives and scope from the outset. The revised sentences are described below.

Page 3, lines 2 to 19 revised version:

“Temporomandibular disorders (TMD), specifically those associated with pain, are currently understood as a set of signs and symptoms that designate a painful musculoskeletal syndrome, often linked to multisystemic changes, including alterations in behavior, emotional status, and social interactions. These manifestations are recognized as expressions of central nervous system dysregulation. Among the main predictors for the development of painful TMD are the presence of comorbidities, non-painful orofacial symptoms (such as self-reported parafunctions), the frequency of somatic symptoms, poor sleep quality, and genetic and epigenetic factors. This study focuses on the painful subtype of TMD, which is of particular clinical relevance due to its impact on quality of life and its association with chronic pain mechanisms [1, 2].

Painful TMD includes diagnoses such as myofascial pain, arthralgia, and headaches attributed to TMD, all of which are characterized by chronic orofacial pain and functional limitations. These painful subtypes differ from intra-articular TMD, which may present with joint noises or movement restrictions without significant painful discomfort. The specificity of diagnosing painful TMD is guided by the Diagnostic Criteria for Temporomandibular Disorders (DC/TMD), which allows for the accurate identification of patients experiencing TMD for the last month. By focusing on painful TMD, this study aims to explore the cognitive-behavioral-emotional factors contributing to the onset and maintenance of pain in this specific population, distinguishing it from the broader spectrum of TMD presentations [3].”

2. The methodology is robust and well-detailed. The statistical analyses and results are presented clearly and concisely. It would be helpful to explicitly mention which variables were excluded due to collinearity during the regression model formation. While this is somewhat implied, providing this information would aid in the reader’s interpretation.

Response: We appreciate the reviewer’s positive feedback on the methodology and the suggestion to improve the clarity of the statistical analyses. In response, we have revised the manuscript to explicitly mention which variables were excluded due to collinearity during the formation of the regression model. Specifically, the final model did not include variables that showed significant collinearity, as indicated by Spearman's correlation coefficient (p < 0.05). This information has been added to the "Statistical Analyses" section to provide readers with a clearer understanding of the model selection process. The added sentences are described below.

Page 8, lines 20 to 31, and page 9, lines 1 to 12

“During the development of the multiple linear regression model, certain variables were excluded due to significant collinearity to maintain the model’s robustness and ensure accurate and reliable estimates. Collinearity was identified using Spearman's correlation coefficient, with variables showing a p-value < 0.05 being excluded from the final model to avoid multicollinearity, which could distort the relationships between the predictors and the outcome. The excluded variables include right upper trapezius pain threshold (OR=0.29, 95%CI=0.07–1.20, p=0.088), right temporomandibular joint (TMJ) pain sensitivity (OR=1.15, 95%CI=0.86–1.54, p=0.346), and left masseter pain threshold (OR=2.28, 95%CI=0.82–6.31, p=0.113). Although these variables showed associations with TMD in the bivariate analysis, their high correlation with other variables in the model necessitated their exclusion. Similarly, sleep quality total score (OR=1.60, 95%CI=0.88–2.93, p=0.124), pain awareness total score (OR=1.07, 95%CI=0.98–1.16, p=0.158), and daytime oral behavior score (OR=1.33, 95%CI=0.88–2.02, p=0.180) were also excluded due to collinearity. Pain vigilance score (OR=1.14, 95%CI=0.95–1.37, p=0.154) significantly correlated with other variables and was similarly removed from the final model.

By excluding these collinear variables, we retained only those predictors that demonstrated no significant collinearity, thus ensuring the independence of the variables in the regression model. The retained variables, including left masseter pain sensitivity, total pain catastrophizing score, difficulty in externalizing feelings, and the distraction facet of the mindfulness scale, provided a more precise and accurate representation of the factors associated with the development of painful TMD. This step was essential for enhancing the model's validity and ensuring that the conclusions drawn are robust and meaningful.”

3. The discussion section is somewhat lengthy and does not thoroughly explore the study’s findings. Previous studies could be summarized more concisely and directly linked to your results.

Response: We appreciate the reviewer’s valuable feedback. In response to this comment, we have revised the discussion section to be more concise and to provide a more explicit focus on the study's findings. We have summarized previous studies more succinctly and emphasized direct connections between our results and the existing literature. Specifically, we have reduced the length of background information and ensured that comparisons with previous research are more clearly tied to the cognitive-behavioral-emotional model explored in our study. Furthermore, we have restructured the discussion to highlight our key findings regarding the role of pain sensitivity, catastrophizing, difficulty in externalizing feelings, and mindfulness distraction in predicting painful TMD. This revision enhances the clarity and relevance of the discussion, ensuring that the findings are thoroughly explored and linked to the broader context of temporomandibular disorder research. We believe these changes address the reviewer’s concern and improve the overall quality of the discussion.

4. I suggest placing greater emphasis on the aspects related to epigenetics, the specificity of your sample, and the extensive number of variables investigated to construct the model—these are significant strengths of your study.

Response: We appreciate the reviewer’s valuable suggestion to highlight the strengths of our study related to epigenetics, sample specificity, and the breadth of variables investigated. In response, we have revised the discussion to emphasize these aspects more. Specifically, we expanded the discussion on the role of epigenetic mechanisms in chronic pain and TMD, underscoring how the dynamic interaction between genetic predisposition and environmental factors may influence pain perception and susceptibility. The use of a monozygotic twin sample is a crucial strength of this study, as it allows us to minimize genetic variability, thus providing a unique opportunity to investigate the influence of non-genetic factors, including cognitive-behavioral-emotional aspects and epigenetic modifications, in the development of painful TMD.

Additionally, we have emphasized the extensive range of variables included in our analysis, which contributes to a more comprehensive understanding of the multifactorial nature of TMD. The breadth of variables investigated in this study, encompassing pain sensitivity, emotional regulation, mindfulness, and catastrophizing, allowed us to develop a robust model that reflects the complexity of the condition. These additions to the discussion underscore the strengths of our methodology and further highlight the significance of our findings.

5. Another aspect that could be more thoroughly discussed is the idea that in genetically identical individuals, other factors may play a more significant role than those typically identified in genetically diverse populations.

Response: We appreciate the reviewer’s thoughtful suggestion and have expanded the discussion to explore how, in genetically identical individuals, non-genetic factors such as environmental, psychological, and epigenetic influences may have a more significant impact than populations with greater genetic diversity. The use of monozygotic twins in our study allows for isolating these non-genetic influences, which provides a unique opportunity to investigate how cognitive-behavioral-emotional factors can override or modulate genetic predispositions. We have added a more detailed discussion on this point, emphasizing the growing evidence that even without genetic variability, psychological factors such as pain catastrophizing, emotional regulation, and mindfulness can play a critical role in pain perception and chronicity. This underscores the importance of focusing on these modifiable factors, which can be targeted in personalized interventions. The sentence below was added to the Discussion Section.

Page 17, lines 15 to 27

“In genetically identical individuals, such as the monozygotic twins in our study, non-genetic factors- environmental, psychological, and epigenetic- emerge as particularly influential in the development of chronic pain conditions like TMD. Our findings reinforce that, even without genetic diversity, cognitive-behavioral-emotional factors such as pain catastrophizing, difficulty in emotional expression, and mindfulness play a pivotal role in pain perception and modulation. Studies have shown that environmental stressors and psychological variables can induce epigenetic changes, influencing gene expression and central sensitization mechanisms [23,24]. This highlights the fact that in genetically identical populations, modifiable factors like emotional regulation and cognitive patterns may have a stronger impact on pain outcomes than genetic predispositions alone. The ability to isolate these influences in monozygotic twins underscores the importance of focusing on psychological and behavioral interventions, which may offer more targeted and effective treatments for chronic pain in individuals with similar genetic backgrounds.”

6. The conclusion could be more concise and more precise.

Response: Thank you for your suggestion. We have revised the conclusion to ensure it is more concise and focused. The revised version now directly summarizes the study's key findings, emphasizing the significant role of cognitive-behavioral-emotional factors in developing painful TMD while highlighting the potential for targeted interventions. We believe this revised conclusion is more straightforward and more aligned with the overall goals of the manuscript.

Minor points

1. *Note: * On page 12, in Table 3, the label for G7 is missing a "T."

Response: Thank you for pointing out this error. We have corrected the label for G7 in Table 3 by adding the missing "T." The updated table now accurately reflects the correct labeling.

Reviewer #2 Manuscript PONE-D-24-27113

General comments

Phenotypes of painful TMD in discordant monozygotic twins according to a cognitive behavioral-emotional model: a case-control study. It was described that the objective of this study “was to investigate variables based on a cognitive-behavioral-emotional model related to the development of painful temporomandibular disorders (TMD) in a sample of monozygotic twins discordant for the condition.”

Response: We would like to thank you for your detailed analysis of our manuscript. Below, we present clarifications of the points raised, and the manuscript has been reformulated based on the suggestions made. We believe the revised version has been improved, and we would like a new manuscript analysis.

Specific points

Please see my comments below:

1. Objective. It is important a clarification why sociodemographic profile, pain sensitivity (pain threshold to pressure, allodynia, and hyperalgesia), oral behaviors, pain vigilance, central sensitization, pain intensity and interference, trigeminal and extratrigeminal pain areas were considered cognitive-behavioral-emotional variables.

Response: Thank you for raising this critical point. Our study categorized these variables as cognitive-behavioral-emotional based on their demonstrated interactions with psychological processes, emotional regulation, and behavioral responses to pain. For example, pain sensitivity, including thresholds for allodynia and hyperalgesia, is modulated by cognitive factors such as pain catastrophizing and emotional states like anxiety and stress, which influence pain perception and pain processing in the central nervous system. Similarly, oral behaviors and pain vigilance reflect learned behavioral responses to pain, influenced by cognitive evaluation of the pain experience and emotional reactions such as fear of pain or hypervigilance.

Central sensitization is a well-established mechanism through which emotional and cognitive factors, like stress or catastrophizing, can exacerbate pain perception by amplifying pain signals. Pain intensity and interference, as well as trigeminal and extra trigeminal pain areas, are closely tied to how individuals cognitively evaluate and emotionally respond to their pain, influencing how they report pain and how it interferes with their daily functioning. Sociodemographic factors, although seemingly unrelated to cognition or behavior, can influence psychological responses to pain based on factors such as socioeconomic status, which may affect stress levels, coping strategies, and access to healthcare, all of which are tied to emotional and cognitive responses to chronic pain. Thus, the inclusion of these variables in the cognitive-behavioral-emotional model is justified based on their interrelated influence on the experience, perception, and modulation of pain, as supported by the biopsychosocial framework of chronic pain.

We have addressed this by revising the introduction, expanding the objectives, and further clarifying why these variables were considered cognitive-behavioral-emotional factors in the discussion. Specifically, we explained their interaction with psychological and emotional processes in modulating pain perception.

2. Method. It is important for the authors to describe the cohort from which the painful TMD patients were recruited, including how many declined participation and how many were excluded.

Response: The participants were not recruited from any specific cohort. Out of the 38 monozygotic twins initially evaluated for the study, 12 pairs met the inclusion criteria for discordant for painful TMD. However, 2 of these pairs declined to participate, leaving a final sample of 10 pairs included in the study. Recruitment was conducted through various media channels (radio, social networks, websites) and promotion via e-mail and messaging applications. This information has been corrected in the manuscript to prevent any potential misunderstandings. We have made the necessary adjustments to ensure that the recruitment process and inclusion criteria are clearly described, avoiding ambiguity. These changes aim to provide greater clarity for readers and ensure that the study's methodology is accurately

---

## [Decision Letter · Decision Letter 1]

26 Nov 2024

PONE-D-24-27113R1Phenotypes of painful TMD in discordant monozygotic twins according to a cognitive-behavioral-emotional model: a case-control studyPLOS ONE

Dear Dr. Leite-Panissi,

Thank you for submitting your manuscript to PLOS ONE. After careful consideration, we feel that it has merit but does not fully meet PLOS ONE’s publication criteria as it currently stands. Therefore, we invite you to submit a revised version of the manuscript that addresses the points raised during the review process.

After reviewing the article, important revisions were identified to enhance the clarity and consistency of the manuscript before it is ready for publication. Key points include the need for language editing, adjustments to the abstract to better clarify the objectives, methodology, and clinical relevance, as well as greater precision in the introduction and conclusion to align them with the results. In the methodology, more detailed information on the recruitment process and clear justifications for exclusions and analyses performed were requested. Additionally, inconsistencies in the statistical approach were highlighted, with a recommendation for review by a statistician to ensure the appropriateness of the model and interpretations.

We look forward to receiving your revised manuscript.

Kind regards,

Cristiano Miranda de Araujo

Academic Editor

PLOS ONE

Reviewers' comments:

Reviewer's Responses to Questions

**Comments to the Author**

1. If the authors have adequately addressed your comments raised in a previous round of review and you feel that this manuscript is now acceptable for publication, you may indicate that here to bypass the “Comments to the Author” section, enter your conflict of interest statement in the “Confidential to Editor” section, and submit your "Accept" recommendation.

Reviewer #2: (No Response)

Reviewer #3: All comments have been addressed

2. Is the manuscript technically sound, and do the data support the conclusions?

Reviewer #2: No

Reviewer #3: Yes

3. Has the statistical analysis been performed appropriately and rigorously? 

Reviewer #2: No

Reviewer #3: N/A

4. Have the authors made all data underlying the findings in their manuscript fully available?

Reviewer #2: Yes

Reviewer #3: Yes

5. Is the manuscript presented in an intelligible fashion and written in standard English?

Reviewer #2: No

Reviewer #3: Yes

6. Review Comments to the Author

Reviewer #2: This study addresses an interesting topic; however, the manuscript requires significant revisions. Please refer to my comments below:

1. English Language: A thorough review and revision of the English language are necessary to improve clarity and readability.

2. Abstract:

a. The abstract lacks clarity. The aim and methodology are not clearly articulated. For instance, it is unclear how the 20 monozygotic twins with discordant conditions were selected.

b. Conclusion and Clinical Relevance are wrong.

c. Presentation: The frequent use of nested parentheses is not appropriate (e.g., (OR=3.29; 95% CI=(0.17–62.8), p=0.428)). Please revise to ensure a cleaner and more professional presentation.

3. Introduction: Please note that self-reported parafunctions do not fall under the category of "orofacial symptoms, the frequency of somatic symptoms, poor sleep quality, and genetic and epigenetic factors." This distinction should be clarified in the introduction.

4. Method:

a. Suggestion: Begin by clearly describing the recruitment process for the study participants. Following this, provide details about the sample: 38 monozygotic twins were initially evaluated, and among them, 12 pairs met the inclusion criteria for discordance in painful TMD.

b. Age Inclusion Criterion: The authors justify including only individuals aged 18–55 to reduce potential age-related confounding factors. While this is a valid rationale, it should also be acknowledged as a limitation. Age could have been accounted for as a potential confounder in the analysis, and the analysis could have been stratified by age to address this concern more comprehensively.

c. Statistical Model Explanation: Upon reviewing the explanation of the statistical model, it appears the authors may not have fully understood the initial comment, as their response does not align with the original concern. If the adjusted model includes variables that are not statistically significant, the authors must explain the implications of these findings, particularly by referencing the 95% confidence intervals as a guide. Again, I strongly suggest consulting a statistician to address these issues thoroughly. Multicollinearity: I disagree with the approach used to assess multicollinearity. Variance Inflation Factor (VIF) should be the primary method, as it evaluates the combined effect of all predictors and provides a direct measure of their impact on the regression model. Correlation analysis can be used as a complementary tool to identify pairwise relationships that may warrant further investigation or justify the use of VIF.

To address potential confounding (line 16, page 7), the authors could adjust the analysis rather than relying on exclusions, which should only be employed if there are very few cases within the excluded group. Additionally, the authors should list the number of subjects excluded in each group.

Statistical Analysis: This section requires verification by a statistician, as it is confusing and contains inaccuracies. For instance, the statement, "The association of each factor with TMD was investigated by fitting pairwise conditional simple logistic linear regression models," is problematic. Is the analysis "simple"? Are "logistic" and "linear" being used interchangeably? Clarify whether collinearity was assessed using logistic regression models and provide an accurate description of the statistical approach. Why was linear regression used (line 28, page 9)? The authors need to specify the dependent variable to justify the choice of this statistical method. The explanation of the four excluded variables is unclear. This section reads more like results than methodology. For example, the finding of a borderline association between trapezius pain threshold and TMD (OR = 0.29, 95% CI = 0.07–1.20, p = 0.088) raises questions—why was this variable excluded despite its potential relevance? The authors should provide a clear rationale for this decision. The sentence, “Although these variables showed associations with TMD in the bivariate analysis, their high correlation with other variables in the model necessitated their exclusion,” is unclear and problematic. Are the authors excluding variables that are associated with TMD? This contradicts the study's objective. I strongly recommend conducting a principal component analysis (PCA) to address multicollinearity and using the resulting components in the logistic regression instead of excluding potentially relevant variables.

5. Results.

a. The sentence, "The participants were not recruited from any Brazilian-specific cohort," should be deleted. Before discussing the evaluation of the 38 monozygotic twins, the authors need to clearly explain the recruitment process of the participants.

b. The statement, “The variables that showed a positive association (according to the OR), however, without statistical significance, were: pain duration…” is incorrect. What is the risk of a Type I error here? Focus on interpreting the 95% confidence intervals (CIs) and p-values.

6. Discussion: The sentence, “Our findings revealed that increased pain sensitivity in the masseter muscle, higher levels of pain catastrophizing, difficulty in externalizing feelings, and higher scores on the distraction facet of mindfulness were significant predictors of painful TMD. These results provide evidence that, even in genetically identical individuals, cognitive and emotional factors play a crucial role in the manifestation of chronic pain associated with TMD” (lines 6–10, page 16), is incorrect and does not align with the study's findings. This discussion must be revised to accurately reflect the results. The claims about significant predictors and their implications should only be made if they are strongly supported by the analysis.

7. Conclusion: The conclusion is not consistent with the findings. The results indicate that the variables analyzed are not significantly associated with TMD and do not suggest any interaction. Instead, the findings highlight that the crude analysis was likely confounded. The conclusion should be rewritten to reflect these points accurately.

Reviewer #3: The introduction aligns well with the research objectives, emphasizing the biopsychosocial model and the complex interactions between genetic, emotional, and behavioral factors. The particularly relevant references to twin studies genetic mechanisms and the role of genetics and epi strengthen the justification for the study design.

The discussion explores how cognitive, behavioral, and emotional dimensions interact with pain perception in TMD, reinforcing the biopsychosocial model. The findings, including the role of pain catastrophizing, emotional regulation, and mindfulness, provide actionable insights for developing targeted interventions.

The study effectively bridges psychosocial theories with clinical practice, providing a multidimensional understanding of TMD. Its innovative design, focusing on monozygotic twins, offers unique insights into genetic and environmental contributions. Future work should aim to validate these findings in diverse and larger populations, using more sophisticated statistical models to unravel the intricate web of influencing factors.

7. PLOS authors have the option to publish the peer review history of their article (what does this mean? ). If published, this will include your full peer review and any attached files.

**Do you want your identity to be public for this peer review?** For information about this choice, including consent withdrawal, please see our Privacy Policy .

Reviewer #2: **Yes: ** Ana Miriam Velly

Reviewer #3: **Yes: ** Stechman-Neto, J

---

## [Author Response · Author response to Decision Letter 2]

26 Dec 2024

Academic Editor – PLOS ONE Manuscript PONE-D-24-27113R1

After reviewing the article, important revisions were identified to enhance the clarity and consistency of the manuscript before it is ready for publication. Key points include the need for language editing, adjustments to the abstract to better clarify the objectives, methodology, and clinical relevance, as well as greater precision in the introduction and conclusion to align them with the results. In the methodology, more detailed information on the recruitment process and clear justifications for exclusions and analyses performed were requested. Additionally, inconsistencies in the statistical approach were highlighted, with a recommendation for review by a statistician to ensure the appropriateness of the model and interpretations.

Reviewer #2 Manuscript PONE-D-24-27113R1

This study addresses an interesting topic; however, the manuscript requires significant revisions.

RESPONSE: Thank you for your feedback and for recognizing the relevance of our study topic. We have carefully reviewed the manuscript and made significant revisions to address the concerns raised. These include improving the clarity of the abstract, refining the methodology section, ensuring consistency between the results and discussion, and revising the language for better readability. We are confident that these changes enhance the quality and scientific rigor of the manuscript.

1. English Language: A thorough review and revision of the English language are necessary to improve clarity and readability.

RESPONSE: Thank you for your observation. A thorough review and revision of the manuscript have been conducted by a native English speaker with expertise in academic writing to ensure clarity and improve readability throughout the text.

2. Abstract:

a. The abstract lacks clarity. The aim and methodology are not clearly articulated. For instance, it is unclear how the 20 monozygotic twins with discordant conditions were selected.

RESPONSE: Thank you for your observation. We have revised the abstract to enhance clarity regarding the aim and methodology. Specifically, we have included a concise description of the recruitment process for the 20 monozygotic twins discordant for painful TMD, detailing the comprehensive strategy employed to identify and include eligible participants. We hope these adjustments address the reviewer’s concerns and improve the overall clarity of the abstract.

b. Conclusion and Clinical Relevance are wrong.

RESPONSE: These topics were rewritten.

“Conclusions: Our study highlights the potential clinical relevance of cognitive-behavioral-emotional variables, such as increased pain sensitivity in the masseter muscle, higher levels of pain catastrophizing, difficulty in externalizing feelings, and higher scores on the distraction facet of mindfulness, in understanding painful TMD. While these variables did not show statistical significance in the adjusted model, their inclusion underscores the importance of exploring these factors in clinical practice. Further research is needed to validate these findings and clarify their role in the development and management of painful TMD.. Clinical Relevance: This study underscores the importance of cognitive-behavioral-emotional factors in the context of painful TMD, suggesting that variables like pain sensitivity and emotional regulation may be valuable for clinical assessment and management strategies. Despite the lack of statistically significant associations, these findings provide a foundation for future research to better understand and address the multidimensional nature of TMD in clinical practice.”

c. Presentation: The frequent use of nested parentheses is not appropriate (e.g., (OR=3.29; 95% CI=(0.17–62.8), p=0.428)). Please revise to ensure a cleaner and more professional presentation.

RESPONSE: Thank you for your observation. We have revised the abstract to remove nested parentheses, ensuring a cleaner and more professional presentation. For instance, "95% CI= (0.17–62.8)" was reformatted to "95% CI: 0.17–62.8." Similar adjustments have been made throughout the text to maintain consistency.

3. Introduction: Please note that self-reported parafunctions do not fall under the category of "orofacial symptoms, the frequency of somatic symptoms, poor sleep quality, and genetic and epigenetic factors." This distinction should be clarified in the introduction.

RESPONSE: The distinction regarding self-reported parafunctions has been clarified in the introduction to ensure alignment with the context of the predictors for painful TMD.

4. Method:

a. Suggestion: Begin by clearly describing the recruitment process for the study participants. Following this, provide details about the sample: 38 monozygotic twins were initially evaluated, and among them, 12 pairs met the inclusion criteria for discordance in painful TMD.

RESPONSE: We fully understand and appreciate your concern regarding the detailed description of the recruitment process, as this was one of the most challenging aspects of conducting this study. During the conceptualization phase, we explored multiple approaches to identify suitable participants. Given the specificity of the condition under investigation (TMD is not widely recognized in the medical field), recruiting monozygotic twins with discordance in painful TMD presented unique challenges. To overcome these difficulties, we implemented a comprehensive recruitment strategy. After obtaining ethical approval, we disseminated information about the study and its inclusion criteria through various channels, including social media platforms, the websites of FORP/USP and FFCLRP/USP, USP Radio and local radio stations in Ribeirão Preto and around, messaging apps targeting student and community groups, and physical posters placed in strategic locations throughout Ribeirão Preto. These efforts resulted in 38 twin pairs contacting our research team, all of whom were scheduled for clinical evaluations. Among these, 12 pairs met the inclusion and exclusion criteria, and 10 pairs agreed to participate in the study. The extended data collection period reflects the significant challenges associated with recruiting such a specific and unique sample. We have incorporated these details into the manuscript to enhance transparency and address your concerns regarding the recruitment process.

b. Age Inclusion Criterion: The authors justify including only individuals aged 18–55 to reduce potential age-related confounding factors. While this is a valid rationale, it should also be acknowledged as a limitation. Age could have been accounted for as a potential confounder in the analysis.

RESPONSE: We agree with this limitation. We have acknowledged this in the discussion, noting that age could have been treated as a potential confounder in the analysis to provide a broader understanding of its influence. This adjustment enhances the transparency and depth of our study's limitations.

c. Statistical Model Explanation: Upon reviewing the explanation of the statistical model, it appears the authors may not have fully understood the initial comment, as their response does not align with the original concern.

RESPONSE: All statistical analyses in this study were conducted by a highly qualified professional statistician with a Master’s and Ph.D. in statistics, specializing in the design and analysis of clinical research. Furthermore, all adjustments to the manuscript and responses to this review have been carefully developed in collaboration with this expert. We greatly value the reviewer’s thoughtful concern regarding the statistical aspects of the study and are diligently working to address and incorporate the insightful suggestions provided. We would also like to respectfully acknowledge the inherent limitations of our study, particularly related to the unique nature of the sample—monozygotic twins discordant for painful TMD—which required the statistical models to be tailored to this specificity. We deeply appreciate the reviewer’s input and are committed to improving the manuscript to ensure it meets the highest standards of scientific rigor.

If the adjusted model includes variables that are not statistically significant, the authors must explain the implications of these findings, particularly by referencing the 95% confidence intervals as a guide. Again, I strongly suggest consulting a statistician to address these issues thoroughly.

RESPONSE: The reviewer is correct to highlight the wide confidence intervals and the high degree of uncertainty they reflect regarding the effect estimates for the variables Pain sensitivity - Face, left, masseter and FFMQ - Distraction Facet Score, as presented in Table 2. We fully acknowledge the uncertainty associated with these effects. However, we believe that these variables may represent important characteristics (signs/symptoms) relevant to clinical practice. It is within this context that we emphasized their discussion in the manuscript, aiming to draw attention to their potential relevance despite the statistical limitations. We appreciate the reviewer’s thoughtful observation and have included additional clarification in the discussion to better address these points.

Multicollinearity: I disagree with the approach used to assess multicollinearity. Variance Inflation Factor (VIF) should be the primary method, as it evaluates the combined effect of all predictors and provides a direct measure of their impact on the regression model. Correlation analysis can be used as a complementary tool to identify pairwise relationships that may warrant further investigation or justify the use of VIF. Clarify whether collinearity was assessed using logistic regression models and provide an accurate description of the statistical approach.

RESPONSE: We acknowledge VIF as a useful measure for detecting multicollinearity. However, we chose to investigate multicollinearity by examining the correlation matrix among the independent variables because we wanted to avoid the effect of overestimating the multiple determination coefficient (R²), given that many independent variables presented p < 0.20 in the bivariate analysis. Furthermore, the literature provides various analogs to R² in the context of logistic regression, which led us to question whether R² would be appropriate in this context. If R² is not appropriate, then VIF = 1/(1-R²) would also not be suitable. Considering this, we opted to examine the correlation matrix among the independent variables, which is also a valid method for investigating multicollinearity.

To address potential confounding (line 16, page 7), the authors could adjust the analysis rather than relying on exclusions, which should only be employed if there are very few cases within the excluded group. Additionally, the authors should list the number of subjects excluded in each group.

RESPONSE: From an analytical perspective, the exclusions (related to interocclusal appliances, alternative therapies, neurological disorders, psychiatric disorders, and pregnancy) were necessary because, when combined, they would generate numerous binary variables, some with low prevalence. Given the sample size, any statistical technique applied would produce effect estimates for these variables with low precision. From a clinical perspective, the exclusions were essential as we aimed to focus on cognitive-behavioral-emotional factors. The number of subjects excluded was included in the Methods section.

Statistical Analysis: This section requires verification by a statistician, as it is confusing and contains inaccuracies. For instance, the statement, "The association of each factor with TMD was investigated by fitting pairwise conditional simple logistic linear regression models," is problematic. Is the analysis "simple"? Are "logistic" and "linear" being used interchangeably?

RESPONSE: Yes. In the bivariate phase, each independent variable is individually associated with the outcome. In statistics, when the outcome variable is associated with a single independent variable through a regression model, this model is referred to as simple regression. Furthermore, since our outcome is binary (presence of TMD / absence of TMD), the regression used was logistic regression. Additionally, logistic regression is one of several regressions within the class of generalized linear models and can therefore be considered linear because the functional relationship of each independent variable with the outcome is assumed to be linear.

Why was linear regression used (line 28, page 9)? The authors need to specify the dependent variable to justify the choice of this statistical method.

RESPONSE: We adopted a linear regression model because we assumed that the functional relationship of each independent variable with our outcome is linear.

The explanation of the four excluded variables is unclear. This section reads more like results than methodology. For example, the finding of a borderline association between trapezius pain threshold and TMD (OR = 0.29, 95% CI = 0.07–1.20, p = 0.088) raises questions—why was this variable excluded despite its potential relevance? The authors should provide a clear rationale for this decision.

RESPONSE: Despite their apparent potential, some variables, including the one mentioned by the reviewer, were excluded due to moderate to high correlation with other equally important variables.

The sentence, “Although these variables showed associations with TMD in the bivariate analysis, their high correlation with other variables in the model necessitated their exclusion,” is unclear and problematic.

RESPONSE: Our justification lies in the pursuit of a parsimonious final model that includes, in its deterministic component, variables with the lowest possible degree of correlation among them.

Are the authors excluding variables that are associated with TMD?

RESPONSE: Due to the small sample size, if all independent variables with p < 0.20 (Table 1) were included in the multiple regression analysis shown in Table 2, the logistic regression model would have extremely low power. For this reason, we used the correlation between the independent variables with p < 0.20 to include in the model presented in Table 2 only the variables that were the most independent from one another.

This contradicts the study's objective.

RESPONSE: We understand that this may seem contradictory, but a good model must also provide precise estimates and be parsimonious. Given the conditions (many independent variables with p < 0.20 eligible for the final model and a small sample size), prioritizing parsimony and selecting the most independent variables was necessary for the screening process.

I strongly recommend conducting a principal component analysis (PCA) to address multicollinearity and using the resulting components in the logistic regression instead of excluding potentially relevant variables.

RESPONSE: The use of PCA would reduce the independent variables to a small set of principal components with a certain degree of orthogonality between them, but it would introduce the disadvantage of interpreting the effects of each principal component on our outcome.

5. Results.

a. The sentence, "The participants were not recruited from any Brazilian-specific cohort," should be deleted. Before discussing the evaluation of the 38 monozygotic twins, the authors need to clearly explain the recruitment process of the participants.

RESPONSE: The sentence was excluded, and the recruitment process was better explained.

b. The statement, “The variables that showed a positive association (according to the OR), however, without statistical significance, were: pain duration…” is incorrect. What is the risk of a Type I error here? Focus on interpreting the 95% confidence intervals (CIs) and p-values.

RESPONSE: Considering the reviewer’s recommendations, the estimation of type I errors for each association is indeed high. For this reason, we recognize that using phrases such as “positive association, but not statistically significant” may seem inconsistent. To address the reviewer’s suggestions, we have revised our interpretation of the assoc

---

## [Decision Letter · Decision Letter 2]

20 Feb 2025

Phenotypes of painful TMD in discordant monozygotic twins according to a cognitive-behavioral-emotional model: a case-control study

PONE-D-24-27113R2

Dear Dr. Leite-Panissi,

We’re pleased to inform you that your manuscript has been judged scientifically suitable for publication and will be formally accepted for publication once it meets all outstanding technical requirements.

Kind regards,

Cristiano Miranda de Araujo

Academic Editor

PLOS ONE

Additional Editor Comments (optional):

Reviewers' comments:

Reviewer's Responses to Questions

**Comments to the Author**

1. If the authors have adequately addressed your comments raised in a previous round of review and you feel that this manuscript is now acceptable for publication, you may indicate that here to bypass the “Comments to the Author” section, enter your conflict of interest statement in the “Confidential to Editor” section, and submit your "Accept" recommendation.

Reviewer #4: All comments have been addressed

2. Is the manuscript technically sound, and do the data support the conclusions?

Reviewer #4: Yes

3. Has the statistical analysis been performed appropriately and rigorously? 

Reviewer #4: Yes

4. Have the authors made all data underlying the findings in their manuscript fully available?

Reviewer #4: Yes

5. Is the manuscript presented in an intelligible fashion and written in standard English?

Reviewer #4: Yes

6. Review Comments to the Author

Reviewer #4: Current and relevant topic. The manuscript is well-structured, and the English is comprehensible. The previous suggestions have been accepted, and no further changes are needed.

7. PLOS authors have the option to publish the peer review history of their article (what does this mean? ). If published, this will include your full peer review and any attached files.

**Do you want your identity to be public for this peer review?** For information about this choice, including consent withdrawal, please see our Privacy Policy .

Reviewer #4: No

---

## [Editor Report · Acceptance letter]

PONE-D-24-27113R2

PLOS ONE

Dear Dr. Leite-Panissi,

I'm pleased to inform you that your manuscript has been deemed suitable for publication in PLOS ONE. Congratulations! Your manuscript is now being handed over to our production team.

Kind regards,

on behalf of

Dr. Cristiano Miranda de Araujo

Academic Editor

PLOS ONE